# MoTE: Mixture of Task Experts for Embedding Models

## Abstract

Dense embeddings are essential for Retrieval-Augmented Generation (RAG), search, classification, and clustering systems. Recent methods improve dense embeddings by enriching them with instructions that incorporate downstream task information, enabling a single model to generate task-specific embedding spaces. However, we empirically show that requiring all tasks to share the same model parameters imposes significant representational limitations. To address these challenges, we introduce Mixture of Task Experts (MoTE), a novel transformer block designed for embedding architectures. MoTE employs dedicated parameter sets tailored to the unique requirements of each task and is paired with a task-aware training framework to improve representation quality. Experiments on 56 datasets spanning 7 tasks demonstrate that MoTE outperforms instruction-conditioned models, achieving, on average, $1.62$ higher NDCG@10 on retrieval datasets, $1.54$ higher MAP on re-ranking datasets, and a $0.65$ improvement in overall performance. Notably, these gains are achieved without altering inference-time information, training data, inference speed, or number of active parameters.

## 1 Introduction

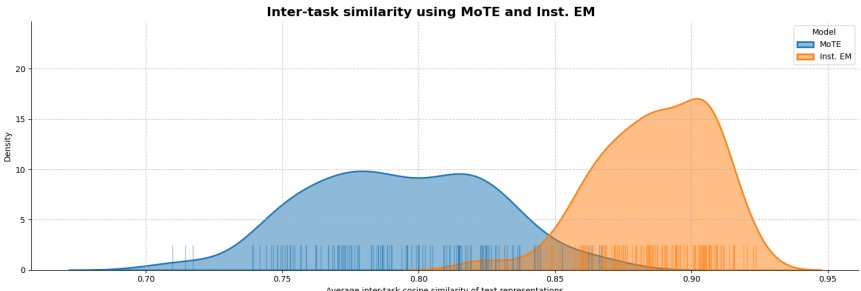

Figure 1: Average inter-task cosine similarity for MoTE and traditional instruction-conditioned embedding models (Inst. EM). After convergence, task representations in Inst. EM are limited to be more similar to each other compared to those produced by MoTE, which optimizes dedicated parameters for each task. This highlights the representational constraints of instruction-based multi-task embedding models.

Semantic text representations are a key component to many real-world applications such as search, recommendation systems, and spam classification. These representations are commonly obtained using a machine learning model that embeds unstructured text into a dense $n$-dimensional vector. Embedding models are developed by simultaneously optimizing the transformer architecture weights towards a wide range of downstream tasks including Retrieval Augmented Generation (RAG) (Gao et al., 2023; Lewis et al., 2020), search (Wise et al., 2020), Semantic Text Similarity (STS) (Chandrasekaran & Mago, 2021), classification (O'Neill et al., 2021; Wang et al., 2021), and clustering (Xu et al., 2015). As a result, a single embedding model must generate high-quality embeddings for an arbitrary range of downstream tasks with varying requirements.

To address these diverging requirements, Wang et al. (2022); Su et al. (2022) propose to compute text representation by first enriching the text with contextual information about its intended usage in the form of a prefix (i.e. instructions) to generate different text representations with the same model. By applying this technique, a single embedding model is used to generate multiple embedding spaces based on the down-stream task that is used to condition the embeddings. For example Wang et al. (2022) uses the instruction "query:" for queries and "passage:" for passages in retrieval tasks. Performance gains derived from theses methods alongside their extensive adoption by state of the art embedding models (Wang et al., 2022; Su et al., 2022) has provided solid evidence of the importance of generating task-aware semantic representations.

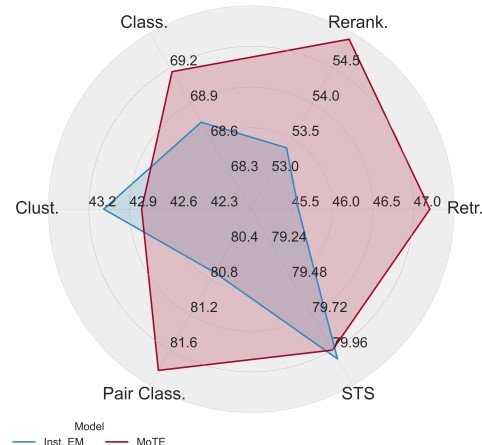

Figure 2: Performance comparison of Inst. EM and MoTE across tasks.

We argue that while existing methods improve the embeddings by incorporating information about their intended use, the use of a single set of weights across tasks limits the task specialization of the resulting representations. This limitation arises because the model's parameters must still be jointly optimized (Figure 1). This limitation becomes even more pronounced during training, where the nuanced requirements of individual tasks are overlooked in favor of selecting a single combination of hyper-parameters[1]. Examples of such hyper-parameters include batching strategies, contrastive loss directionality, and contrastive temperature. This paper addresses these limitations through the following contributions:

1. We introduce *Mixture of Task Experts (MoTE)*[2], a novel Mixture of Experts (MoE) transformer block for embedding models with a dedicated set of parameters for each task. MoTE constitutes an alternative to traditional multi-task contrastive training of dense transformer blocks and can be easily up-cycled from contemporary instruction embedding models.

2. We introduce a novel *Task-Aware Training Strategy* that captures the nuanced requirements of different tasks by enabling a flexible hyper-parameter configuration during training.

Experiments on 56 datasets across 7 tasks (Muennighoff et al., 2022) show that MoTE achieves, on average, $1.62$ higher NDCG@10 on retrieval datasets, $1.54$ higher Mean Average Precision (MAP) on re-ranking datasets and $0.65$ higher performance across datasets (Figure 2) compared to traditional multi-task contrastive training. Critically, these improvements are achieved with identical input information, training data, latency, and number of active parameters. Lastly, we show that expert averaging can mitigate MoTE's increased memory footprint while partially preserving its performance improvements.

## 2 BACKGROUND

### 2.1 TEXT EMBEDDING MODELS

Embedding models are generally trained with a data-quality curriculum consisting of three sequential stages of decreasing data volume and increasing quality: generic pre-training, contrastive pre-training, and contrastive fine-tuning. Historically, embedding models (Gao et al., 2021) have leveraged encoder-only models such as BERT (Devlin et al., 2018) or RoBerta (Liu et al., 2019) during the generic pre-training stage. However, with the development of more powerful Large Language Models (LLMs) (Muennighoff et al., 2024; Touvron et al., 2023a;b), recent research suggests to

---

[1]In this paper, we use the term "hyper-parameter" broadly to encompass various training configurations, such as choice of loss function, selection of negative examples, batching strategy, and more.

[2]In this paper we use MoTE to refer to both the transformer block and the resulting architecture for readability purposes.

re-use these existing decoder-only models to initialize the weights of the contrastive pre-training stage[3]. Following the generic pre-training stage, the contrastive pre-training stage (Wang et al., 2022; Su et al., 2022; Nussbaum et al., 2024; Chen et al., 2024) leverages curated co-occurring text pairs naturally found in the web such as question-answer pairs from Reddit[4] and StackOverflow[5], or title-passage pairs from Wikipedia[6] pages. During this stage the model is trained with contrastive objectives such as the InfoNCE (Oord et al., 2018) using in-batch negatives (Chen et al., 2020; He et al., 2020). Finally, the model goes through a contrastive fine-tuning stage (Wang et al., 2022; Su et al., 2022; Nussbaum et al., 2024; Chen et al., 2024) that leverages a smaller amount of curated datasets - like MSMARCO (Bajaj et al., 2016) or Natural Questions (NQ) (Kwiatkowski et al., 2019) - with an anchor, a positive, and a reduced number of high-quality hard negatives (negatives that are hard to distinguish from the positive) per sample[7].

## 2.2 INSTRUCTION EMBEDDINGS

A core challenge during the development of embedding models is the careful calibration of a unique embedding space capable of powering multiple downstream applications without significant performance degradations in any of them (Muennighoff et al., 2022; Neelakantan et al., 2022). For example, a STS task requires the semantically similar questions "Who was Isaac Newton?" and "Who was the father of Calculus?" to be close in the embedding space while a retrieval task requires them to be far apart because the latter does not answer the former (Gao et al., 2021). To address this challenge, Wang et al. (2024); Muennighoff et al. (2024); Wang et al. (2023); Zhang et al. (2023) augment the input text with contextual information about the downstream task. This strategy leverages text instructions to condition the generation of different embedding representations using the same model and text. Contemporary methods leveraging this strategy vary in the level of detail provided in the instructions. From example, Wang et al. (2022) use text-type or task-specific information such as "Query:" and "Document:" in retrieval use cases while Su et al. (2022) use dataset-specific instructions with more detailed information such as "Represent the Amazon comment for classifying the sentence as positive or negative:".

## 2.3 MIXTURE-OF-EXPERTS

The application of MoE (Yuksel et al., 2012) to generative models (Jiang et al., 2023; Touvron et al., 2023b) has recently increased the popularity of MoE transformer architectures (Jiang et al., 2024; Gao et al., 2022b). MoE architectures are composed of MoE blocks that replace traditional dense blocks throughout the architecture or at every other block. In dense transformer blocks all intermediate token representations are processed through a single Multi-Layer Perceptron (MLP). In contrast, MoE blocks leverage a dynamic routing mechanism $g$ and multiple MLPs (or experts) to process intermediate token representations. Specifically, for each intermediate token representation, the routing mechanism $g$ uses the token representation to identify the subset of experts that should process the representation (Lepikhin et al., 2020). Replicas of the intermediate token representation are then dispatched to the selected experts where they are processed independently. Lastly, a pooling layer aggregates the experts' output to generate a unique representation for each of the tokens. During training, an auxiliary expert-balancing objective is leveraged to learn the routing mechanism while ensure a consistent training of and a balanced workload across the experts modules (Jiang et al., 2024; Gao et al., 2022b).

---

[3]While embedding models initialized from decoder-only models (Muennighoff et al., 2024; Touvron et al., 2023a;b) perform better than those initialized from encoder-only models (Wang et al., 2022; Su et al., 2022; Nussbaum et al., 2024), they also tend to be larger (e.g. ∼7B parameters vs 100M - 1.5B parameters respectively). Thus, incurring in higher latency, lower throughput and higher output memory footprint.

[4]https://www.reddit.com/

[5]https://stackexchange.com/

[6]https://www.wikimedia.org/

[7]When hard negatives are not included in the datasets, existing literature proposes a range of approaches to mine them from the corpora: random sampling, semantic-similarity search, manual labeling/review, etc.

### 2.4 MODEL WEIGHT AVERAGING

Model weight averaging (McMahan et al., 2017; Gao et al., 2022a; Matena & Raffel, 2022; Wortsman et al., 2022) is a simple yet effective model combining technique where several model checkpoints of the same architecture are combined by averaging their model parameters. This has been shown to improve performance at inference time (Popel & Bojar, 2018), as well as generalization to out-of-distribution inputs (Rame et al., 2022). Similarly to the expert averaging described in section 5, Chronopoulou et al. (2023) shows that weight averaging of adapters trained on different domains yields better performance in hold-out domains compared to single adapter routing.

## 3 METHOD

### 3.1 MIXTURE OF TASK-EXPERTS

We set ourselves to tackle the representational limitations of instruction-conditioned embedding models while retaining their ability to share relevant task-independent knowledge. Following with the example in Section 2.2, consider that we want to embed "Who was Isaac Newton?", once for a retrieval-query use case and once for a classification use case. The embedding model should possess the ability to learn and apply specialized transformations that can prioritize information that will aid in matching relevant documents (e.g., the question asks about the entity Issac Newton and "who was" is tailored to retrieve biographical details) or emphasizing global features of the sentence for the classification tasks (e.g., question type classification) according to the intended downstream task. Second, in contrast to fully independent specialized models, task-independent knowledge should be shared and accessible across tasks. For example that "Newton" in this context refers to a person instead of a unit of measure should be learned across all experts regardless of which task the training data contained physics examples. This section describes how MoTE mitigates this limitations (Wang et al., 2024; Zhang et al., 2023).

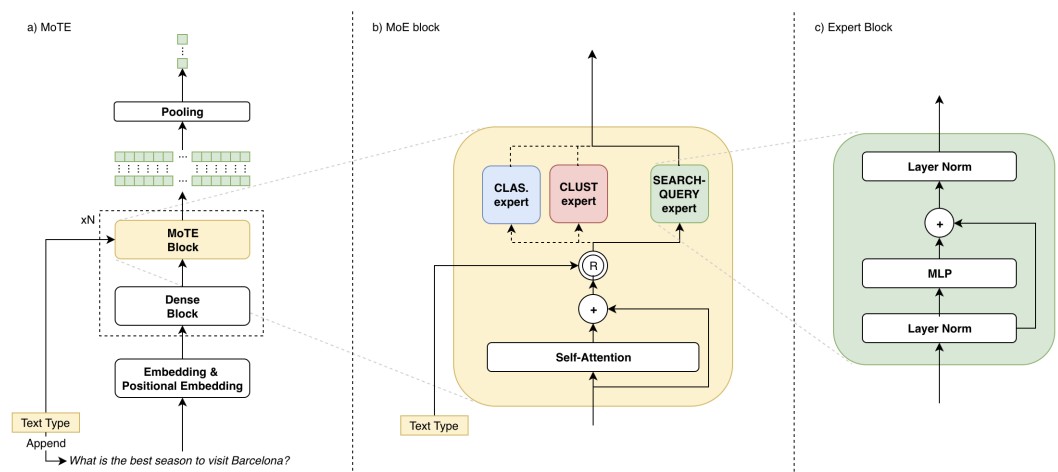

Figure 3: MoTE leverage global and single-task parameters to balance the shared and specialized task knowledge respectively. Subfigure (a) illustrates the MoTE architecture with MoE blocks incorporated every other transformer block, subfigure (b) illustrates the routing mechanism within a MoE block and subfigure (c) illustrates the components of a single expert block.

MoTE accepts tuples $(i, t)$ where $i \in \mathcal{I}$ represents the task instruction providing contextual information about the downstream task (e.g., "Query ", "Passage: ") and $t$ denotes the input text. Consistent with contemporary instruction embedding models, the concatenated text $[i|t]$ is processed through the embedding layer and transformer blocks. In contrast, the task instruction $i$ is also used by the MoE blocks (Figure 3-a).

The MoTE architecture interleaves MoTE blocks at Every Other Block (EOB) of the transformer architecture[8]. The MoE blocks employ a novel context-based routing mechanism $R : \mathcal{I} \to \mathcal{E}$ to establish a mapping between the tasks $i \in \mathcal{I}$ and the expert modules $e \in \mathcal{E}$ (Figure 3-b) that is used to route the intermediate representation of $[i|t]$ through it's task-associated expert. Task-routing offers several advantages over standard MoE token-level routing. First, tokens belonging to the same sequence are deterministically routed through the same expert based on their associated task. This approach facilitates the model's capacity to learn task-specific experts. Second, token-level MoE requires of an auxiliary objective to ensure that the tokens are routed across the experts in a balanced fashion to ensure expert specialization while prevent expert overloading or collapse. In contrast, task-routing mitigates this complexity by optimizing intermediate token representations only through the primary loss function and relying on contextual task information to determine the sequence routing. Critically, by utilizing only one expert per MoE block, MoTE maintains the same number of active parameters, latency and computational efficiency as the original dense transformer architecture while enabling task-specialized weight representations.

As illustrated in Figure 3-c, the expert blocks $e$ contain a MLP block and two normalization layers. The MLP component facilitates specialized transformations for different text types while the normalization layers enables MoTE to learn different centroids and variances for each taskBa (2016).

### 3.1.1 UP-CYCLING INITIALIZATION

MoTE's task-routing mechanism requires of task-augmented inputs to train the task experts. However, contemporary embedding models only leverage task-augmented inputs during contrastive training stages while relying on a task-agnostic generic pre-training stage. Consequently, this presents a challenge because experts can neither be meaningfully trained through the generic pre-training stage nor dense transformers checkpoints resulting from the generic pre-training stage can be directly leveraged to initialize the MoTE architecture prior the contrastive pre-training stage[9].

To overcome this limitation, we propose to up-cycle (Komatsuzaki et al., 2022; He et al., 2024) the dense transformer checkpoint resulting from the generic pre-training stage to initialize MoTE's contrastive pre-training stage. Existing dense embedding models are typically trained using task-agnostic self-supervised objectives such as masked language modeling or next token prediction. Consequently, their checkpoints contain a single MLP that is incompatible with MoTE's multi-expert architecture. Our proposed initialization strategy involves consistently replicating the original MLP block from the dense checkpoint across $|\mathcal{E}|$ expert modules.

By up-cycling pre-trained dense transformer checkpoints, we achieve four critical objectives. First, MoTE is capable of preserving the foundational linguistic knowledge acquired during generic pre-training. Second, it creates a coherent initialization of the task-experts when composed with the remaining weights of the architecture. Third, it enables us to leverage the rich pool of existing pre-trained langauge models. Lastly, it constitutes a flexible initialization strategy that supports a varying range of task-specialized experts in MoTE. This approach bridges the gap between task-agnostic generic pre-training and task-routed embedding models while providing a novel application of the MoE up-cycling mechanism.

### 3.2 TASK-AWARE TRAINING

Embedding models must generate semantically meaningful representations across a diverse range of tasks and domains. We argue that downstream tasks exhibit distinct representation requirements, which manifest in nuanced approaches to hyper-parameter selection and training strategies. Consider the divergent characteristics of clustering and STS tasks. Clustering applications require identifying similar data points while distinguishing them from arbitrarily different samples across domains. Consequently, high-quality hard negatives must exhibit a large domain diversity independently of the anchor. In contrast, STS tasks require to discriminate between semantically similar yet distinct textual representations. Consequently, high-quality hard negatives must exhibit high se-

---

[8]Note that our choice of interleaving MoTE blocks at EOB is arbitrary and not fixed. In practice, MoTE blocks provide practitioners with the flexibility to determine which dense blocks to up-cycle based on their specific memory and performance requirements. This trade-off is further explored in Appendix C.

[9]Note that this challenge is not present between contrastive pre-training and contrastive fine-tuning as both stages leverage task information and the same number of experts

mantic similarity to the anchor. These contrasting requirements often create optimization challenges where strategies optimal for one task can degrade the performance in another.

To overcome this limitation we propose to combine MoTE with task-aware contrastive training. This method constructs task-specific mini-batches and dynamically adjusts training hyper-parameters — such as batching strategies, contrastive temperature, and objective functions — based on each task's requirements. It employs task-specific hyper-parameters and a hierarchical data structure that organizes training samples by task and source dataset, ensuring that each expert learns from data relevant to its task during contrastive learning. Task metadata is passed from the dataloaders to the training module to facilitate this process. When combined with MoTE, this approach enables gradient specialization during training, allowing each expert to be optimize using task-specific hyper-parameters and data effectively.

A drawback of this approach is that it can increase the search space by a factor of $|\mathcal{T}|$, where $|\mathcal{T}|$ is the number of tasks. To address this, we propose two mitigation strategies: first, selectively focus hyperparameter tuning on a subset of parameters relevant to tasks where optimal training strategies are unclear or where preliminary analysis indicates significant performance variability. This reduces the multiplicative factor in the search space by applying it only to this subset of parameters, rather than the full parameter set. Second, utilize the dense embedding model as a proxy to systematically analyze hyperparameter impacts across tasks (Appendix A). This strategy reduces the search space to be equivalent to training the dense transformer model plus one additional optimization run that dynamically combines task-specific adjustments. Together, these strategies effectively mitigate the computational overhead of task-aware optimization, making it comparable to traditional hyper-parameter tuning methods.

### 3.3 EXPERT AVERAGING

Due to the nature of the MoTE architectures we can increase the expressive power of the embedding model without increasing the dimensionality of the embeddings or incurring latency overheads. However, MoTE can reduce the model's throughput in selective cases where we do not have enough memory to process large batch sizes — like document classification or retrieval indexing — due to the increased GPU memory consumed by the specialized weights.

To address this we developed a post-training approach that decreases the memory footprint by combining the independently-trained experts into a single expert. We refer to the resulting model as Expert Averaged (EA). Specifically, we obtain EA by averaging each of the layers across experts resulting in a single MLP block. After conducting this process we obtain a model that retains over 98.2% of the MoTE performance across tasks while maintaining the same number of layers and memory footprint as the original dense transformer architecture (Section 5.5).

## 4 EXPERIMENTAL METHODOLOGY

Evaluations are conducted on the (Nussbaum et al., 2024; Wang et al., 2022; 2023; Chen et al., 2024; Su et al., 2022) Massive Text Embedding Benchmark (MTEB) (Muennighoff et al., 2022) comprising 7 tasks and 56 datasets. We leverage the same metrics and evaluation procedure reported in MTEB and report average dataset performance for each task and across all tasks in the experiment. Table 1 provides a summary of the dataset distribution and task metrics.

| Task | Number of Datasets | Metric |
|---|---|---|
| Retrieval | 15 | NDCG@10 |
| Classification | 12 | Accuracy |
| Clustering | 11 | Validity Measure |
| Re-ranking | 4 | Mean Average Precision (MAP) |
| Pair Classification | 3 | Average Precision (AP) |
| STS | 10 | Spearman correlation |
| Summarization | 1 | Spearman correlation |

Table 1: Metrics and distribution of datasets across tasks in the MTEB benchmark

To ensure experimental reproducibility, we employ the open-source artifacts from Nussbaum et al. (2024). Specifically, all candidates are trained using their publicly available neural architecture[10], initialized with the pre-trained masked language modeling (MLM) checkpoint[11], and trained on identical contrastive pre-training datasets for classification, clustering, and retrieval tasks[12]. These datasets were preprocessed using top-2 consistency filtering as proposed by (Nussbaum et al., 2024; Wang et al., 2022). Throughout the contrastive training process we consistently utilize the InfoNCE loss (Oord et al., 2018) and a AdamW optimizer with learning rate of $5 \times 10^{-6}$ and weight decay of $0.1$. The model is trained with a batch size of $6,144$ for a single epoch. During training we maintain an identical dataset-to-task mapping[13] as Nussbaum et al. (2024). Unless otherwise specified, the remaining hyper-parameters[14] are consistently replicated from Nussbaum et al. (2024) to ensure experimental control and isolate the independent variable in each experiment.

MoTE is implemented with 4-experts associated with classification, clustering, retrieval-query and retrieval-document. Each of the experts is associated with instructions "classification: ", "clustering: ", "search query: ", and "search document: " respectively. Throughout our experiments we comparatively evaluates MoTE against instruction-conditioned Embedding Models trained on identical datasets and using the same inference information. Both candidates are trained using task-aware training leveraging the contrastive temperature and batching strategy found in Appendix A using Inst. EM.

To further investigate the performance characteristics of MoTE, we incorporate three Specialized Embedding Models (SEM) into our comparative analysis. The SEM models are implemented using dense transformer architectures, each trained independently on distinct datasets focused on classification, clustering, and retrieval tasks. In contrast to MoTE, SEM models possess a larger aggregate parameter count, with the critical distinction that none of their layers undergo joint training.

## 5 RESULTS

### 5.1 PERFORMANCE ACROSS TASKS

This section explores the performance of MoTE in tasks seen during training, namely retrieval, classification and re-ranking. We compare MoTE, Inst. EM and SEM trained using the same configuration. For SEM, each task column value reflects the score of the task-specialized model. Thus, the only difference between the Inst. EM and MoTE candidates is the architecture.

| | Retrieval (NDCG@10) | Classification (accuracy) | Clustering (validity measure) | Average (Dataset performance) |
|---|---|---|---|---|
| SEM | 33.45 | 64.08 | 39.11 | 45.55 |
| Inst. EM | 45.58 | 68.74 | **43.09** | 52.47 |
| MoTE | **47.20** | **69.17** | 42.81 | **53.06** |

Table 2: Performance comparison of SEM (task-specialized models trained only on datasets for their respective tasks), Inst. EM, and MoTE across tasks seen during training. While Inst. EM and MoTE share the same training data and inference-time information, they differ in architecture, with MoTE utilizing task-specialized experts.

Aligned with trends in multi-task embedding training, the results show that using a single model across tasks (Inst. EM and MoTE) achieves superior performance compared to task-specialized models (SEM). Furthermore, MoTE demonstrates additional gains over instruction-conditioned em-

---

[10]Available at: `https://github.com/nomic-ai/contrastors/blob/main/src/contrastors/models/encoder/modeling_nomic_bert.py`

[11]Accessible at: `https://huggingface.co/nomic-ai/nomic-bert-2048`

[12]Dataset source: `https://huggingface.co/datasets/sentence-transformers/embedding-training-data`

[13]Configuration available at: `https://github.com/nomic-ai/contrastors/blob/main/src/contrastors/configs/data/contrastive_pretrain.yaml`

[14]Available at: `https://github.com/nomic-ai/contrastors/blob/main/src/contrastors/configs/train/contrastive_pretrain.yaml`

bedding models (Inst. EM), achieving $+1.62$ points in retrieval tasks and $+0.59$ points on average, highlighting the benefits of task-specialized experts (Table 2).

## 5.2 GENERALIZABILITY TO NEW TASKS

Conditioning embedding models on specific instructions to tailor text representations for a finite set of tasks risks creating overly specialized embedding spaces. This risk is further accentuated by MoTE, which leverages specialized parameters for each task. In this section, we evaluate MoTE generalizability to unseen tasks including re-ranking, pair-classification, STS, and summarization. To map new tasks to appropriate specialized models, instructions, or experts, we follow the instruction-task mapping strategy outlined in Nussbaum et al. (2024). Specifically, we associate retrieval task with re-ranking tasks and classification tasks with pair classification, STS and summarization tasks across all candidates.

| | Reranking | Pair Classification | STS | Summarization | Average |
|---|---|---|---|---|---|
| | (MAP) | (AP) | (Spearman corr.) | (Spearman corr.) | (Dataset perf.) |
| SEM | 48.03 | 73.30 | 75.09 | 26.45 | 55.72 |
| Inst. EM | 53.37 | 80.72 | **80.02** | 33.76 | 61.97 |
| MoTE | **54.91** | **81.83** | 79.96 | **33.97** | **62.67** |

Table 3: Performance comparison of Specialized Embedding Models (SEM), Instruction-conditioned Embedding Models (Inst. EM), and MoTE on unseen downstream tasks

Similar to Inst. EM, MoTE demonstrates strong generalization capabilities to unseen tasks (Table 3). Notably, MoTE achieves a $+1.54$ MAP improvement for re-ranking tasks, a $+1.11$ AP improvement in AP for pair classification, and only minor performance drops in STS. Overall, MoTE outperforms Inst. EM by $+0.7$ points on average across datasets of unseen tasks, highlighting its ability to generalize while maintaining robust performance.

## 5.3 LIMITATIONS OF INSTRUCTION-BASED MULTI-TASK LEARNING

Our initial conjecture was that relying solely on instructions to generate task-specific embedding spaces restricts the degree of task specialization these spaces can achieve. To test this hypothesis, we compare the inter-task similarity of embedding spaces produced by MoTE and Inst. EM, trained under identical conditions. Specifically, we embed a corpus of 128 Wikipedia articles[15] as classification, clustering, retrieval corpus, and retrieval query representations.

For each element in the corpus, we measure the cosine similarity between task-specific representations and compare the average similarity across tasks for MoTE and Inst. EM. Statistical significance is assessed using a one-sided Welch's t-test based on the hypothesis formulated in Expression 1, where $\mu_{\mathcal{M}}^{(T_1, T_2)}$ represents the true cosine similarity between tasks $T_1$ and $T_2$ for model $\mathcal{M}$.

$$H_0 : \mu_{MoTE}^{(T_1, T_2)} = \mu_{\text{Inst. EM}}^{(T_1, T_2)}$$
$$H_A : \mu_{MoTE}^{(T_1, T_2)} < \mu_{\text{Inst. EM}}^{(T_1, T_2)} \tag{1}$$

Figure 4 illustrates the distributional differences in inter-task similarity between MoTE and Inst. EM. In addition to visual distinctions across most task pairs, the p-value in most cases falls below the $0.05$ significance threshold, allowing us to reject the null hypothesis that MoTE and instruction-conditioned embeddings achieve, on average, the same level of task independence. These findings support our initial hypothesis that instruction-conditioned models with shared weights constrain the representational capacity of embedding models.

---

[15]Dataset available at: `https://huggingface.co/datasets/sentence-transformers/embedding-training-data/resolve/main/SimpleWiki.jsonl.gz`

Interestingly, we observe that clustering and document retrieval experts do not exhibit a statistically significant increase in task specialization. This suggests that their embedding spaces may have a lower degree of independence, indicating potential for further memory optimization by sharing the same expert with different instructions for clustering and retrieval tasks. We leave this investigation for future work.

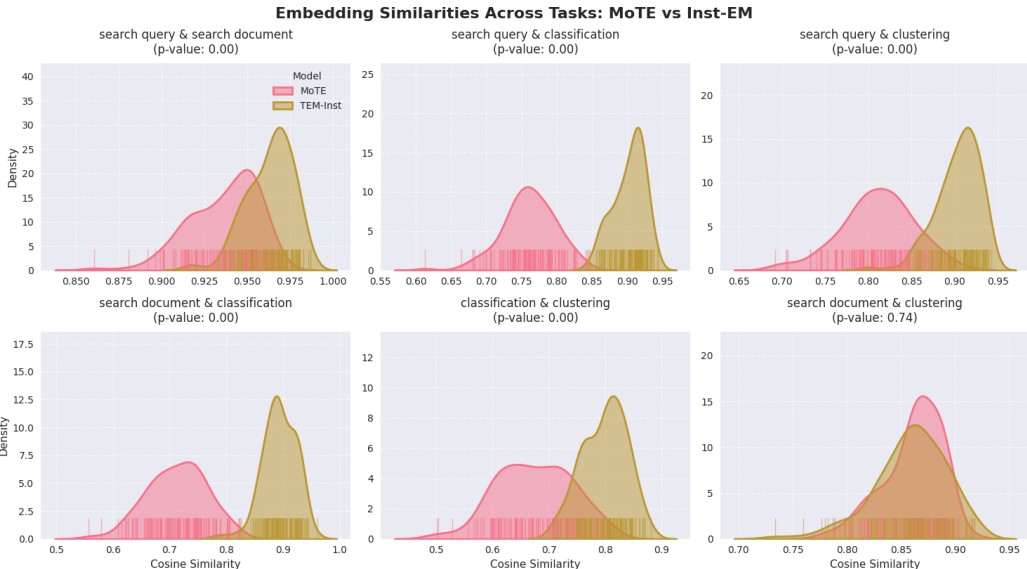

Figure 4: Inter-task similarity of text representation when using Inst. EM and MoTE. We observe that in most cases MoTE is able to reach a higher degree independence across tasks after the multi-task contrastive learning stage.

## 5.4 ABLATION: EFFECT OF THE TASK-AWARE TRAINING STRATEGY

We leverage task-aware training instead of traditional task-agnostic training. Task-aware training capture the distinctive training nuances across task by creates a dependency between the current mini-batch's task and the training hyper-parameters. In contrast, traditional task-agnostic training leverages the hyper-parameter combination that leads to the overall higher performance, disregarding individual task requirements. In this experiments we ablate both experiments over an hyper-parameter space composed of the batching strategy and contrastive temperature dimensions selected using the analysis in Appendix A over MTEB. Specifically, task-aware training employs homogeneous batching for retrieval mini-batches and heterogeneous batching in classification and clustering mini-batches and a contrastive temperature of $0.03$ for retrieval and classification mini-batches and $0.06$ for clustering mini-batches. Static training leverages an heterogeneous batching strategy and a $0.03$ contrastive temperature.

|  | Static training | Task-aware training |
|---|---|---|
| Retrieval[1] | 46.48 | **47.20** |
| Classification[2] | 69.00 | **69.17** |
| Clustering[3] | **42.82** | 42.81 |
| Re-ranking[4] | 54.69 | **54.91** |
| Pair Classification[5] | **81.93** | 81.83 |
| STS[6] | 79.70 | **79.96** |
| Summarization[6] | **34.06** | 33.97 |
| Average dataset performance | 58.78 | **59.07** |

[1] NDCG@10;  [2] accuracy;  [3] validity measure;  [4] MAP;  [5] AP;  [6] Spearman correlation

Table 4: Comparison of MoTE when trained with a static or a dynamic regime.

Table 4 show that task-aware training leads to an average dataset performance improvement of $0.29$ largely driven by improvements in retrieval of $0.71$ NDCG@10 with only minnor regressions ($< .1$) in clustering, pair classification and STS.

## 5.5 EA: EXPERT AVERAGING

The main drawback of MoTE when compared to current dense alternatives is its higher memory consumption which can lead to lower throughput in real world applications such as RAG-indexing or classification which require bulk inference. To alleviate this problem we introduced an Expert Averaging (EA) compression mechanism to the MoTE architecture which alleviated the increased memory footprint at no additional training cost by averaging the expert blocks (Section 3.3). In this section we systematically study the performance retention of this technique compared to both MoTE and Inst. EM.

|  | Inst. EM | MoTE | EA |
|---|---|---|---|
| Retrieval[1] | 45.58 | 47.20 | 46.93 |
| Classification[2] | 68.74 | 69.17 | 68.82 |
| Clustering[3] | 43.09 | 42.81 | 42.70 |
| Reranking[4] | 53.37 | 54.91 | 54.91 |
| Pair Classification[5] | 80.72 | 81.83 | 81.59 |
| STS[6] | 80.02 | 79.96 | 79.13 |
| Summarization[6] | 33.76 | 33.97 | 33.38 |
| Average dataset performance | 58.43 | 59.07 | 58.60 |

[1] NDCG@10;     [2] accuracy;     [3] validity measure;     [4] MAP;     [5] AP;     [6] Spearman correlation

Table 5: We observe that the EA-compressed MoTE retains $0.17$ average dataset performance improvement compared to Inst. EM.

EA mitigates MoTE's increased memory footprint while retaining $0.17$ average dataset performance over Inst. EM. The average performance retention is largely driven by retrieval and re-ranking tasks with of $1.35$ NDCG@10 and $1.54$ MAP performance improvements of EA over Inst. EM, respectively.

## 6 CONCLUSION

Traditionally, embedding models have relied on a single model to produce a single embedding space. The recent rise of instruction-conditioning strategies challenges this paradigm by enabling the generation of distinct downstream vector representations for the same input text using contextual task instructions. However, as demonstrated in this paper, this approach limits the model's expressive power. To address this, we introduced MoTE, a framework that employs a dedicated set of parameters for each task to enhance task specialization.

Through task-aware training, MoTE effectively captures the nuances of individual tasks during multi-task learning, resulting in significant performance improvements. Specifically, MoTE achieves, on average, $1.62$ points higher NDCG@10 on retrieval datasets, $1.54$ points higher Mean Average Precision (MAP) on re-ranking datasets, and $0.65$ higher dataset performance across tasks. Critically, these improvements are achieved without altering inference-time information, training data, latency, or the number of active parameters, demonstrating MoTE's efficiency and effectiveness compared to traditional instruction-conditioned embedding models.

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

# Appendices

## A  HYPERPARAMETER SENSITIVITY ANALYSIS

### A.1  BATCHING STRATEGY

The selection of in-batch negative samples is inherently task-dependent, with varying embedding characteristics influencing optimal sampling approaches. Tasks emphasizing *local* semantic nuances, such as Semantic Textual Similarity (STS) (Agirre et al., 2012), require negatives that capture proximal semantic distinctions. Consider paraphrase identification, where sentences like "Vivendi shares closed 1.9 percent at 15.80 euros in Paris after falling 3.6 percent on Monday" and "in new york, vivendi shares were 1.4 percent down at $18.29" demand fine-grained contextual differentiation. Conversely, tasks focused on *global* semantic representations, including classification and clustering, benefit from more diverse negative sampling strategies that capture broader semantic variations.

To empirically validate this hypothesis, we conducted a controlled experiment training an embedding model under two distinct sampling regimes: homogeneous and heterogeneous. By maintaining all other experimental parameters constant, we isolated the impact of sampling strategy.

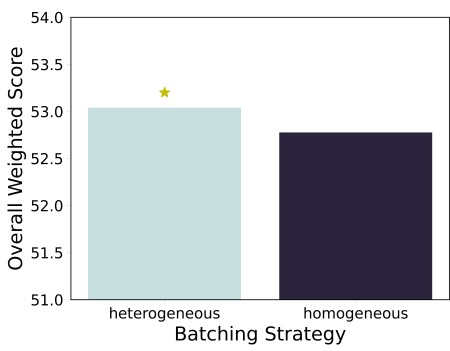

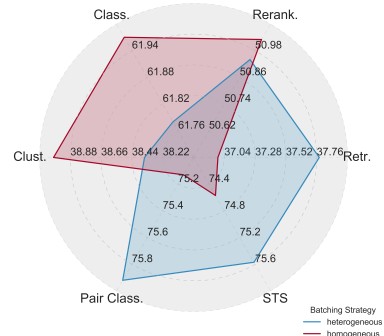

Figure 5: Aggregate Performance: Batching Strategy

Figure 6: Task-Specific Performance: Batching Strategy

Figure 5 demonstrates that heterogeneous sampling yields the highest aggregate performance metric. However, a granular analysis in Figure 6 reveals nuanced task-specific variations: homogeneous sampling optimizes performance for local semantic tasks, while heterogeneous sampling proves superior for global embedding objectives. This underscores the critical insight that a universal sampling strategy cannot uniformly optimize performance across diverse downstream tasks.

## A.2 CONTRASTIVE TEMPERATURE

Similar to batching strategies, different contrastive temperatures capture different requirements across downstream task. Lower contrastive temperatures place a higher relative weight on samples that are more semantically similar to the anchor, thus capturing *local* semantic nuances. In contrast, higher temperatures provide a more uniform distribution of the negative weights to capture difference across a more diverse set of negatives.

To empirically validate this hypothesis, we conducted a controlled experiment training an embedding model under two distinct contrastive temperatures: $0.03$ and $0.06$. By maintaining all other experimental parameters constant, we isolated the impact of contrastive temperature.

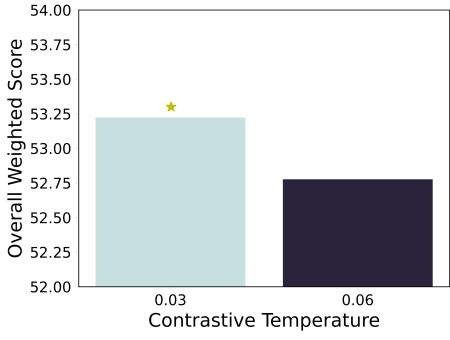

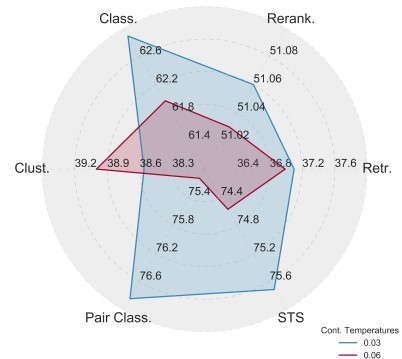

Figure 7: Overall optimal contrastive temperature.

Figure 8: Optimal contrastive temperature per downstream task.

Figure 7 show that the overall optimal configuration is to choose a temperature of 0.03 but a more detailed analysis in Figure 8 shows that this configuration, while beneficial to the overall performance, hurts clustering tasks.

# B   ABLATION: LEARNED TOKEN-LEVEL VS DETERMINISTIC SEQUENCE-LEVEL ROUTING

MoE literature (Section 2.3) leverages context-aware token representations to route the token through different experts allowing for a better generalization across domains. In contrast, MoTE generates dense embeddings by routing examples based on their task (sequence-level) without enforcing additional expert-balancing constraints. We systematically compare the performance difference impact of leveraging sequence-level task-routing mechanism instead of the traditional token-level routing mechanism (TLR) with the same number of experts. The input for both candidates is the augmented original text with the task prefix ("classification: ", "clustering: ", "search query: ", "search document: ").

| | Token-level routing | Sequence-level routing |
|---|---|---|
| Retrieval[1] | 45.17 | **47.20** |
| Classification[2] | 67.98 | **69.17** |
| Clustering[3] | 42.69 | **42.81** |
| Reranking[4] | 54.40 | **54.91** |
| Pair Classification[5] | 80.84 | **81.83** |
| STS[6] | 78.47 | **79.96** |
| Summarization[6] | 32.67 | **33.97** |
| Average dataset performance | 57.86 | **59.07** |

[1] NDCG@10;    [2] accuracy;    [3] validity measure;    [4] MAP;    [5] AP;    [6] Spearman correlation

Table 6: Comparison between token-level routing and task-based sequence-level routing (SLR)

Table 6 show that directly leveraging the task context (SLR) as an expert-routing mechanism leads to performance gains when compared to indirectly leveraging this information through the intermediate token representations (TLR).

# C   ABLATION: MOE ARCHITECTURE DESIGN

Contemporary MoE literature proposes two designs to integrated experts in the transformers architectures balancing between local and global layers. Specifically, MoE layers can be implemented at EOB (Lepikhin et al., 2020) or at Every transformer Block (EB) (Fedus et al., 2022). In this experiment we ablate this design choice for MoTE when implemented with task-experts at EOB and at EB.

| | MoE at Every Block | MoE at Every Other Block |
|---|---|---|
| Retrieval[1] | **47.52** | 47.20 |
| Classification[2] | 69.16 | **69.17** |
| Clustering[3] | **42.89** | 42.81 |
| Re-ranking[4] | **54.93** | 54.91 |
| Pair Classification[5] | 81.81 | **81.83** |
| STS[6] | **80.06** | 79.96 |
| Summarization[6] | **34.32** | 33.97 |
| Average dataset performance | **59.19** | 59.07 |

[1] NDCG@10;    [2] accuracy;    [3] validity measure;    [4] MAP;    [5] AP;    [6] Spearman correlation

Table 7: Comparison between architectural MoE designs: experts Every-Other Block (EOB)

Table 7 shows that using MoE at EB and at EOB leads to 59.19 and 59.07 average dataset performance, respectively. Additionally, the memory footprint for these configurations is measured at 2.2 GB for EB and 1.6 GB for EOB. Given the significant memory savings, we implement MoTE with experts at EOB at the expense of 0.12 average dataset performance.

To further analyze memory consumption, we evaluate its scaling with the number of experts for Nussbaum et al. (2024)'s neural architecture. For EOB, the memory usage increases as follows: 1 expert consumes 1.1 GB, 2 experts require 1.3 GB, 3 experts need 1.5 GB, and 4 experts use 1.6 GB. In contrast, EB demonstrates a steeper memory growth, starting at 1.1 GB for 1 expert and rising to 1.5 GB, 1.8 GB, and 2.2 GB for 2, 3, and 4 experts, respectively.

