# OpenReview forum: "MoTE: Mixture of Task Experts for Embedding Models"
_ICLR.cc/2025/Conference — Submitted to ICLR 2025_

### Official Review · Reviewer_MkSg · 2024-10-30

**Soundness:** 2
**Presentation:** 2
**Contribution:** 3
**Rating:** 5
**Confidence:** 4

**Summary:**

This paper introduces a novel architecture, MoTE, which allocates a dedicated set of parameters for each task to enhance dense text embeddings. MoTE offers a robust approach to improving dense text embeddings by employing a mixture of experts. This architecture, combined with a dynamic training strategy and WAVE, achieves better task specialization without intensively increasing latency or memory overhead. Experiments were conducted on the MTEB benchmark to compare MoTE with previous instruction-based embedding model training approaches and analyze the contributions of its techniques.

**Strengths:**

1. This paper explores the novel Mixture of Task Experts architecture for embedding models, which has the potential to effectively address the challenges posed by various retrieval tasks with differing intents.
2. It conducts numerous analytical experiments and investigates the impact of training configurations, yielding insights that may aid further research on MoTE.

**Weaknesses:**

1. The experiments on MTEB do not compare MoTE against mainstream retrieval models, such as E5 and BGE, nor does it provide comparisons by using training data of mainstream retrieval models, which weakens the reliability of the experimental results.
2. Many analytical experiments only present results without in-depth analysis, such as in sections 4.5 and 4.6.
3. The writing is relatively poor. For example, the two paragraphs in section 3.3 appear to be repetitive.

**Questions:**

1. Is the WAVE method introduced in section 4.7 too simplistic, merely averaging the expert layers post-training but can maintain the performance?
2. As many current MoE architectures are based on LLMs, could this approach be more suitable for retrieval models grounded on LLMs?

---

> ### Author Response · Authors · 2024-11-22
>
> Thank you for your thoughtful and detailed review. We appreciate your recognition of the novel Mixture of Task Experts (MoTE) architecture and the insights provided by our experimental analysis.
>
> #### Weaknesses
>
> 1. We appreciate your concern about the choice of baseline models. Our rationale for selecting Nussbaum et al. (https://arxiv.org/pdf/2406.18587v1) as our primary baseline is three folded. First, reproducibility - Nussbaum et al. provide intermediate checkpoints, hyper-parameters, architecture, pre-processed data, and publicly available code. Given that MoTE reuse most of these artifacts, this baseline eases the reproducibility of our results. Second, relevance to production settings - smaller models, like the one from Nussbaum et al., are more practical in production due to their lower latency, inference cost and vector storage cost. Demonstrating MoTE’s benefits in this context ensures a more tight alignment with real-world deployment scenarios. Third, performance ranking - after both contrastive training phases, the Nussbaum et al. model achieves a top-6 ranking in its size category, performing better than E5 but slightly below BGE. This ensures it remains a competitive baseline for our experiments.
>
> 2. Thank you for this suggestion. We have expanded *Sections 4.5* and *4.6* (now part of the Appendix) to include a deeper analysis alongside memory analysis. Additionally, we have included similarity analysis across the task embedding space to provide readers more insight into both the current limitation of instruction-conditioned embedding models and how MoTE overcomes them.
>
> 3. We acknowledge the writing issue and appreciate your feedback. We have reviewed the entire manuscript to improve its presentations and readability.
>
> #### Questions
>
> 1. We do not find the efficacy of the expert averaging (what we referred as WAVE in the initial submission) method surprising, as it aligns conceptually with established model averaging techniques, which are known to enhance performance and generalization capabilities (e.g., https://arxiv.org/pdf/2302.07027,  https://arxiv.org/pdf/2205.09739). To provide additional context, we have added a background section on model averaging (*Section 2.4*) in the revised manuscript, as also suggested by another reviewer.
>
> 2. This is an excellent point, and we appreciate your insight. While our current experiments focus on encoder embedding models, we agree that exploring MoTE’s applicability to LLM-based retrieval models and hope that this paper encourages future research in this direction.
>
> Thank you again for your constructive feedback. We hope these updates address your concerns and provide additional clarity. We are happy to further refine the paper based on additional suggestions.

---

> > ### Comment · Reviewer_MkSg · 2024-11-26
> >
> > Thank you for addressing my questions and concerns. I'm willing to raise my rating from 3 to 5.

---

> > > ### Author Response · Authors · 2024-11-26
> > >
> > > Thank you for revisiting our work and for your willingness to adjust your rating based on our updates. We sincerely appreciate your constructive feedback, which has significantly improved the quality of the manuscript.
> > >
> > > We truly value your thoughtful comments and believe the additional feedback can make a stronger case for MoTE’s contribution to embedding model research and its applicability to real-world scenarios. If there are any remaining concerns or suggestions for further improvement, we would be grateful to address them.
> > >
> > > Thank you again for your time and consideration.

---

### Official Review · Reviewer_Yimc · 2024-10-31

**Soundness:** 2
**Presentation:** 2
**Contribution:** 2
**Rating:** 3
**Confidence:** 4

**Summary:**

This paper tries to introduce the MOE architecture into the general embedding models and provide different hyperparameters for each task expert to improve performance. Meanwhile, the Weight Average of Vector Experts is obtained by merging the trained experts, which reduces the overall number of parameters and keeps the performance almost equal to that before merging.

**Strengths:**

1. The improvements presented in this paper are straightforward, with expected gains from the MOE architecture and Weight Averaging.
2. The ablation experiments are sufficient, covering performance variations across different settings.
3. The final model maintains the backbone parameter count, ensuring a fair comparison.

**Weaknesses:**

1. The work lacks innovation. It primarily enhances model performance by applying two established strategies (MOE and Weight Averaging) to generic representations without offering new insights.
2. The paper centres on technical implementation but (1) does not commit to open-sourcing the code and (2) provides a very brief methodology, omitting details such as the implementation of the Task-Aware Training Strategy.
3. The paper appears incomplete, falling short of the 10-page limit and lacking a thorough methodology section. Grammar and citation issues also affect readability.
4. Weight Averaging is a model merging method, yet no background on model merging is provided, making the work hard to follow for readers unfamiliar with this area.

**Questions:**

1. The performance difference between "TEM-Inst." and MoTE is minimal (less than 1% in Tables 1 and 2). However, as "TEM-Inst." results don’t appear to be from a public source and the hyperparameter search process isn’t clearly detailed, could you clarify the hyperparameter tuning efforts made to ensure a fair comparison?

2. The statement that “existing methods, ..., limits the degree of task-specialization of the final vector representation” is unclear. In Tables 1 and 2, "TEM-Inst." clearly outperforms STEM, suggesting that multi-task joint learning generally enhances performance across tasks. Thus, setting aside the results in MTEB:

(1) What evidence shows that the trained representation space is task-specialized?

(2) To what extent does the representation space differ between tasks?

(3) Why can’t instructions alone resolve this issue?

---

> ### Author Response · Authors · 2024-11-22
>
> Thank you for your thoughtful review, feedback and for highlighting the strengths of our paper.
>
> #### Weaknesses
>
> 1. We would like to clarify that our goal is not to invent new MoE architectures, but to show that adaptations of MoE can be leveraged to address the limitations of instruction-conditioned embedding models, which leads to clear performance improvements as demonstrated in our results. Besides, while MoE has been widely adopted in LLMs, it is rare to see them applied to embedding models and has not been previously used to address the multi-task aspect of such models. Although this approach may not be feasible for LLM applications, it fits naturally into the training and evaluation paradigm of embedding models. By applying MoE in this context, we offer a fresh perspective on the interplay between MoE, instruction-conditioned embedding models and multi-task learning. However, we acknowledge that this contribution could have been more clearly articulated in the original submission. To mitigate this weakness we clarified the area of our contributions in the *Introduction*.
>
> 2. Regarding weaknesses *(2)* - *(4)*: Thank you for the detailed feedback. We mitigated weakness *(2)* in the new review by expanding the *Methods* section to provide a detailed description of the implementation of task-aware training, up-cycling and MoTE. However, if any aspect remains unclear, we are happy to iterate further based on your suggestions. Regarding weakness *(4)* - We acknowledge the lack of foundational background on weight averaging and based on your suggestion have included a dedicated subsection in the revised manuscript. Lastly, the expanded methodology and background section, alongside a detailed analysis in task similarity using MoTE and traditional instruction-conditioned model have expanded the manuscript to the $10$ page limit with $3$ and $2$ additional pages for reference and appendix, respectively (addressing weakness *(3)*).
>
> #### Questions
>
> 1. Thank you for pointing this out. We have clarified our hyper-parameter selection process in the *Experimental Methodology* section. Specifically, we use hyper-parameters from Nussbaum et al (https://arxiv.org/pdf/2406.18587v1) with a batch size of 6144 to ease reproduction. We only ablate the contrastive temperature and batching strategy parameters using their base transformer architecture with static training.
>
> 2. These are great questions. Thanks for asking. The specialized weights in MoTE are trained with data specific to each task, optimizing representations inherently for those tasks. We included *Section 5.3* to compare to what extent the embedding spaces differ between tasks for both instruction-conditioned embedding models and MoTE. We observe that in most cases the task spaces produced by instruction-conditioned embedding models exhibit a statistically higher inter-task similarity compared to MoTE (Figure $4$). This demonstrates the existing limitations of optimizing a single set of weights for all tasks and highlights how MoTE achieves higher performance by relaxing this constraint while maintaining the same number of active parameters per task. To our surprise this was not the case for clustering and retrieval documents for which our analysis shows that those task spaces hold a higher similarity (Figure $4$ - bottom, right). Thus, we believe that MoTE could be further memory optimizations by using one expert and different instructions for retrieval documents and clustering. We leave this line of research for future work.
>
> Thank you again for your constructive feedback, and we hope these updates and explanations address your concerns.

---

> > ### Author Response · Authors · 2024-11-26
> >
> > As the discussion period is nearing its end, we wanted to kindly follow up and ensure that our responses have adequately addressed your questions and concerns. Based on your valuable feedback, we have made several key updates to the paper, including:
> >
> > 1. Expanding the methodology section to provide detailed descriptions of task-aware training and weight averaging, as well as addressing the implementation details you highlighted.
> > 2. Including a dedicated subsection on the background of weight averaging, as suggested.
> > 3. Adding task similarity analyses to demonstrate the specialized representation spaces achieved by MoTE and clarifying the limitations of instruction-conditioned embedding models.
> >
> > We deeply appreciate your thoughtful comments and suggestions, which have significantly improved the clarity and quality of our work. If there are any additional points you'd like us to address or clarify, we would be more than happy to do so.
> >
> > Thank you once again for your constructive feedback, and we look forward to your final thoughts.

---

> > > ### Comment · Reviewer_Yimc · 2024-11-26
> > >
> > > Thanks to the authors' feedback, I think the current completion of the paper is much improved compared to the initial version.
> > >
> > > However, I think this work still falls short of an ICLR paper because it does not have a single exciting innovation on the theoretical, empirical, or technical level. Specifically, although the authors are the first to introduce MOE to text embedding tasks, I did not gain some new insights from the paper, nor did I find that SOTA effects can be achieved using only MOE.
> > >
> > > Therefore, I chose to maintain my score, and I recommend that the authors consider other conferences such as ACL or SIGIR.

---

### Official Review · Reviewer_161q · 2024-11-06

**Soundness:** 3
**Presentation:** 4
**Contribution:** 3
**Rating:** 6
**Confidence:** 4

**Summary:**

This paper presents a novel approach for task-specific fine-tuning of embedding models in a mixture-of-experts framework, termed Mixture of Task Experts (MoTE). The MoTE model generates task-conditioned embeddings through dedicated parameter sets for each task, allowing for specialized representations tailored to specific downstream requirements. To address the increased memory demands associated with this approach, the authors introduce Weight Average of Vector Experts (WAVE), which mitigates the computational overhead of MoTE by efficiently managing memory without sacrificing performance. This dual approach overcomes the limitations of single-expert models applied across diverse tasks.

To evaluate its effectiveness, the proposed method was tested on seven in-scope and out-of-scope tasks across 56 datasets, demonstrating notable performance gains due to the MoTE architecture.

**Strengths:**

- This work presents innovative methods for generating task-aware embeddings, building on and extending previous research.
- It introduces a novel approach that leverages task-specific hyperparameters to create specialized embeddings while effectively managing the memory consumption challenges of task-conditioned models.
- The method demonstrates practical applicability by addressing the limitations of using a single expert across multiple downstream tasks, showing promise for real-world implementations.
- The paper is clearly written, with well-chosen examples and motivating illustrations that enhance comprehension.
- It includes a robust experimental setup, testing across seven tasks (both in-scope and out-of-scope) with a total of 56 datasets.
- A thorough ablation study and in-depth analysis strengthen the insights derived from the experimental results.

**Weaknesses:**

- This work represents an incremental advancement in mixture-of-expert models, focusing on generating task-aware embeddings for downstream tasks and addressing limitations of prior approaches.
- The approach shows only marginal gains for out-of-scope tasks, such as in pair similarity. For task-aware applications like Classification, Clustering, and Retrieval, in general, they share semantic or latent spaces across them. However, the applicability of the proposed method does not generalize effectively to out-of-scope tasks and domain?
For example, in Table 2, the proposed approach yields only slight improvements on semantic textual similarity (STS) tasks, highlighting limited performance gains in out-of-scope settings.

**Questions:**

NA

---

> ### Author Response · Authors · 2024-11-22
>
> Thank you for your thorough review and constructive feedback. We appreciate your recognition of the innovative methods, practical applicability, and strong experimental setup in our paper.
>
> #### Weaknesses
>
> 1. While we acknowledge that our work builds upon existing mixture-of-expert (MoE) architectures, we believe it represents a significant advancement in applying MoE to embedding models. Specifically, MoTE introduces task-specific embeddings with clear improvements in task-aware applications, addressing key limitations of prior approaches. However, we recognize that this contribution could have been more clearly articulated in the original submission. To address this, we have expanded our analysis in the revised draft to better contextualize the limitations of prior methods and demonstrate how MoTE mitigates these challenges. These additional studies further highlight MoTE's ability to effectively enhance embedding model expressiveness and performance across diverse downstream tasks.
>
> 2. We think there might have been some misunderstandings regarding Table 2 (now Table 3 in the revised draft). MoTE, our proposed method, in fact consistently demonstrates improvements across the majority of tasks, including task-aware applications such as classification, clustering, and retrieval. While the semantic textual similarity (STS) task is the one out-of-scope task where MoTE shows only marginal gains, it is important to note that MoTE outperforms prior methods, including TEM-Inst, on all other tasks. We hope this clarification resolves any confusion and highlights the broad applicability of our approach.
>
>
> Please let us know if we understood your concerns correctly and if the above explanation and attached review addresses them. Thank you once again for your insightful feedback and for helping us improve the clarity and rigor of our work.

---

> > ### Author Response · Authors · 2024-11-26
> >
> > As the discussion period is drawing to a close, we would like to kindly follow up to ensure that our responses have addressed your feedback and clarified the concerns raised in your review. Based on your valuable input, we have made several key updates and adjustments to the manuscript, including:
> >
> > 1. Expanding the contextual analysis to better articulate MoTE’s contributions relative to existing multi-task embedding models, as well as its effectiveness in addressing the limitations of prior methods.
> > 2. Providing a more detailed explanation of Table 2 (now Table 3) to clarify MoTE’s consistent improvements across a variety of tasks, while also acknowledging its marginal degradation on out-of-scope tasks like semantic textual similarity (STS).
> >
> > We sincerely appreciate your thoughtful review and suggestions, which have significantly improved the depth and clarity of our work. If there are any further concerns or points requiring clarification, we would be happy to address them.
> >
> > Thank you again for your constructive feedback and for helping enhance the quality of our paper. We look forward to hearing your final thoughts.

---

### Official Review · Reviewer_ezqd · 2024-11-10

**Soundness:** 3
**Presentation:** 3
**Contribution:** 3
**Rating:** 5
**Confidence:** 4

**Summary:**

The paper introduces a novel approach for embedding models called Mixture of Task Experts (MoTE) that incorporates task-specific MLP blocks within the transformer architecture. During training, different tasks are directed to task-specific blocks, allow the model to generate different embeddings for the same query based on the task context.
This adds new parameters (the task-specific blocks) corresponding to the number of tasks during training. To reduce the parameter count, they apply WAVE (Weight Average of Vector Experts) where they average the weights of the task specific blocks.

**Strengths:**

The embeddings of two different queries may need to be close or far apart depending on the task. Their approach to generating task-specific embeddings enables the optimization of distinct parameters for each task, promoting better task specialization.

**Weaknesses:**

1. Adding a new ML block after every other transformer layer will add significant memory requirement. It would be beneficial to explore optimization strategies to reduce this, such as adding ML blocks only in the topmost layers.
2. The authors show that using the WAVE approach to reduce the gpu requirement retains 98.2% average performance. However, the more important metric is how much of the improvement is retained. Table 6 indicates that WAVE does not provide significant performance improvement.
3. There should be an analysis of GPU memory increment as the number of tasks during training increases.
4. From Table 3, task-aware training does not seem to improvement performance for most tasks.

Minor:
1. There are duplicate lines in Section 3.3.
2. Page 5 line 216: texty-type should be text-type

**Questions:**

N/A

---

> ### Author Response · Authors · 2024-11-22
>
> Thank you for your detailed and insightful review. We greatly appreciate your recognition of the strengths of our approach and your constructive feedback on areas that could be improved. We have carefully addressed the concerns you raised and made corresponding updates to the paper:
>
>
> #### Weaknesses
>
> 1. You are absolutely correct that the inclusion of task-specialized experts introduces an inherent memory cost, even as we maintain low inference latency. We acknowledge that our initial draft could have explained this trade-off more clearly. To address this:
>    1. We have clarified in the Introduction that Expert Average (EA) (previously referred to as WAVE) represents an initial step toward mitigating memory requirements.
>
>     2. As you noted, the decision to upcycle experts in the EOB configuration was arbitrary. We have added a paragraph in the Methods section explaining that practitioners have the flexibility to customize what dense blocks to up-cycle into MoTE blocks —whether in all layers, every other layer, the topmost layer, etc.— based on their memory and performance requirements.
>     3. While EA establishes a baseline for balancing memory and performance, we acknowledge that more advanced memory optimization techniques (e.g., reusing experts across tasks) are promising directions for future research. We have elaborated on this in Section 5.3.
>
> 2. We revised the discussion of EA to better communicate its role and impact. EA is primarily intended as a baseline for reducing memory overhead while retaining a significant portion of the performance gains achieved by MoTE. Specifically, EA achieves 58.60 average performance points compared to the 59.07 points of MoE with respect to the 58.43 points of the base instruction-conditioned model while maintaining the same memory footprint.
>
>     1. In response to your feedback, we added a row to Table 5 showing the average performance per dataset for clarity and removed the percentage retention metric to avoid potential confusion.
>
> 3. As suggested, we have included a GPU memory consumption analysis as a function of the number of tasks (experts) during training. This analysis is provided in Appendix C, comparing both the EB and EOB configurations.
>
> 4. We think there might have been a misunderstanding here. While task-aware training yields an average improvement of +0.29 over the baseline, we recognize that this trend was not sufficiently highlighted in the original draft. We have updated Table 4 to include the average performance across datasets and expanded the results analysis paragraph with a more detailed explanation, making this improvement more evident.
>
> Additionally, we also corrected the paper writing to improve clarity and comprehension. We hope that these revisions address your concerns and clarify the contributions and significance of our work. Thank you again for your valuable feedback, which has helped improve the quality of our paper.

---

> ### Author Response · Authors · 2024-11-26
>
> As the discussion period nears its conclusion, we would like to follow up to ensure that our responses have satisfactorily addressed your feedback and clarified the concerns raised in your review. Based on your insightful comments, we have made several key updates and improvements to the manuscript:
>
> 1. We clarified in the Introduction that Expert Average (EA), formerly referred to as WAVE, is an initial strategy to balance memory and performance. We also expanded the Methods section to outline how practitioners can flexibly decide which dense blocks to up-cycle into task-specific experts, providing options to balance the performance and memory footprint trade-off.
>
> 2. As suggested, we conducted an analysis of GPU memory consumption as the number of tasks increases during training. This detailed analysis is now included in Appendix C, covering both EB (Every Blocks) and EOB (Every-Other Block) configurations.
>
> 3. We revised the discussion of EA’s role to highlight its utility in significantly reducing memory overhead while retaining substantial performance gains. To improve clarity, we added dataset-specific averages to Table 5 and removed the confusing metric that you kindly highlighted.
>
> 4. We addressed concerns regarding task-aware training by explicitly emphasizing its average improvement of +0.29 points over the baseline. Table 4 now includes average performance across datasets, and we expanded the accompanying discussion to make the trend more evident.
>
> 5. We corrected duplicated lines in Section 3.3, addressed minor errors (e.g., "texty-type"), and improved clarity throughout the paper.
>
> Thank you for recognizing the strengths of our approach and for providing such constructive feedback. If there are any additional concerns or areas for further clarification, we would be more than happy to address them.
>
> We sincerely appreciate your time and effort in helping us enhance the quality of our work. We look forward to hearing your final thoughts.

---

> > ### Comment · Reviewer_ezqd · 2024-11-26
> >
> > Thank you for addressing the concerns I had. The revisions have improved the paper, and I am raising my rating from 3 to 5.
> > That said, the performance gains achieved are modest. I believe there is still room for further refinement of the approach to achieve more substantial improvements.

---

### Meta-Review · Area_Chair_ccvc · 2024-12-19

**Metareview:**

This paper proposes a Mixture of Task Experts (MoTE) architecture for embedding models that allocates dedicated parameters for different tasks to improve dense text embeddings. The approach combines task-specific experts with a dynamic training strategy and weight averaging to achieve better task specialization without significantly increasing latency or memory overhead. Experiments on 56 datasets spanning multiple tasks demonstrate modest improvements over instruction-conditioned baselines.

The reviewers acknowledged some strengths of the work, including its well-conducted experiments (reviewers 161q and MkSg) and practical applicability to real-world scenarios (reviewer 161q). However, there were several major concerns. For example, reviewer Yimc pointed out that the work lacks innovation, primarily applying established techniques (MOE and Weight Averaging) without offering new insights. Multiple reviewers (ezqd and Yimc) noted that the performance gains were minimal - less than 1% improvement over the baseline in many cases. Reviewer ezqd raised concerns about memory requirements and the effectiveness of the weight averaging approach in retaining improvements.

Given these considerations and the average rating of 4.75, the paper is recommended for rejection.

**Additional Comments On Reviewer Discussion:**

During the discussion period, the authors made substantial efforts to address the above concerns by clarifying the hyperparameter selection process, adding detailed memory consumption analysis, expanding the methodology section and background on weight averaging, and including new analysis of task similarity and embedding spaces. While these revisions improved the paper's clarity and technical depth, they did not fully address the core concerns about limited innovation and modest performance gains.

---

### Decision · Program_Chairs · 2025-01-22

Reject